# Applications of Blocker Nucleic Acids and Non-Metazoan PCR Improves the Discovery of the Eukaryotic Microbiome in Ticks

**DOI:** 10.3390/microorganisms9051051

**Published:** 2021-05-13

**Authors:** Yurie Taya, Gohta Kinoshita, Wessam Mohamed Ahmed Mohamed, Mohamed Abdallah Mohamed Moustafa, Shohei Ogata, Elisha Chatanga, Yuma Ohari, Kodai Kusakisako, Keita Matsuno, Nariaki Nonaka, Ryo Nakao

**Affiliations:** 1Laboratory of Parasitology, Department of Disease Control, Faculty of Veterinary Medicine, Graduate School of Infectious Diseases, Hokkaido University, N 18 W 9, Kita-ku, Sapporo 060-0818, Japan; yurie1103@vetmed.hokudai.ac.jp (Y.T.); g.kiono115@gmail.com (G.K.); wessam@czc.hokudai.ac.jp (W.M.A.M.); m.abdallah@vetmed.hokudai.ac.jp (M.A.M.M.); s.ogata@vetmed.hokudai.ac.jp (S.O.); chatanga@vetmed.hokudai.ac.jp (E.C.); y_ohari@vetmed.hokudai.ac.jp (Y.O.); kusaki@vmas.kitasato-u.ac.jp (K.K.); nnonaka@vetmed.hokudai.ac.jp (N.N.); 2Ecological Genetics Laboratory, National Institute of Genetics, Mishima, Shizuoka 411-8540, Japan; 3Department of Animal Medicine, Faculty of Veterinary Medicine, South Valley University, Qena 83523, Egypt; 4Department of Veterinary Medicine, Lilongwe University of Agriculture and Natural Resources, Lilongwe 219, Malawi; 5Laboratory of Veterinary Parasitology, School of Veterinary Medicine, Kitasato University, Towada, Aomori 034-0818, Japan; 6Division of Risk Analysis and Management, International Institute for Zoonosis Control, Hokkaido University, N 20 W 10, Kita-ku, Sapporo 001-0020, Japan; matsuk@czc.hokudai.ac.jp; 7International Collaboration Unit, Research Center for Zoonosis Control, Hokkaido University, N 20 W 10, Kita-ku, Sapporo 001-0020, Japan; 8One Health Research Center, Hokkaido University, N 18 W 9, Kita-ku, Sapporo 060-0818, Japan

**Keywords:** artificial nucleic acid, eukaryotic microbiome, next-generation sequencing, protists, tick

## Abstract

Ticks serve as important vectors of a variety of pathogens. Recently, the viral and prokaryotic microbiomes in ticks have been explored using next-generation sequencing to understand the physiology of ticks and their interactions with pathogens. However, analyses of eukaryotic communities in ticks are limited, owing to the lack of suitable methods. In this study, we developed new methods to selectively amplify microeukaryote genes in tick-derived DNA by blocking the amplification of the 18S rRNA gene of ticks using artificial nucleic acids: peptide nucleic acids (PNAs) and locked nucleic acids (LNAs). In addition, another PCR using non-metazoan primers, referred to as UNonMet-PCR, was performed for comparison. We performed each PCR using tick-derived DNA and sequenced the amplicons using the Illumina MiSeq platform. Almost all sequences obtained by conventional PCR were derived from ticks, whereas the proportion of microeukaryotic reads and alpha diversity increased upon using the newly developed method. Additionally, the PNA- or LNA-based methods were suitable for paneukaryotic analyses, whereas the UNonMet-PCR method was particularly sensitive to fungi. The newly described methods enable analyses of the eukaryotic microbiome in ticks. We expect the application of these methods to improve our understanding of the tick microbiome.

## 1. Introduction

Ticks (Acari, Ixodida) are important vectors of various pathogens, including viruses, bacteria, and protozoa, infecting both humans and animals. Ticks also carry nonpathogenic microorganisms, some of which are in a symbiotic relationship with ticks, for instance, through nutritional support [1,2]. Furthermore, several lines of evidence suggest that the pathogenic bacterial burden in ticks is influenced by the presence of other nonpathogenic bacteria [3,4]. Thus, knowledge of the tick microbiome is of great importance for understanding the physiology of ticks and their interactions with pathogens.

Some arthropods harbor eukaryotic microorganisms that are in a symbiotic relationship with them and their associated prokaryotes. For example, lower termites harbor a variety of symbionts of cellulolytic flagellates in their guts [5,6,7]. These flagellates harbor symbiotic bacteria either intracellularly or on their surface, contributing to cellulose degradation and motility in the flagellum [6,8,9]. In addition, it has been observed that some symbiotic bacteria involved in amino acid synthesis have been replaced with fungal symbionts derived from entomopathogenic fungi in cicadas [10]. Considering these examples, eukaryotic taxa cannot be neglected in analyses of the arthropod microbiome. However, little is known about microeukaryotes in ticks. Asides from the well-known tick-borne phylum Apicomplexa, including *Babesia* and *Theileria*, only a few organisms, including trypanosomes [11,12], fungi [13,14], and nematodes [15], have been reported in ticks.

The tick prokaryotic microbiome has been investigated by polymerase chain reaction (PCR) amplification of the prokaryotic 16S rRNA gene (rDNA) with universal primers and subsequent high-throughput sequencing with next-generation sequencing (NGS) technologies [16,17,18]. The tick eukaryotic microbiome is yet to be extensively explored [19] because tick 18S rDNA, whose copy number is higher than that of other eukaryotes in DNA samples extracted from ticks, is over-amplified by conventional PCR with universal primers. Hence, it is necessary to develop a new method to suppress the amplification of tick sequences while simultaneously allowing the amplification of the sequences of other eukaryotes. The addition of artificial nucleic acids such as peptide nucleic acids (PNAs) can block the amplification of dominant sequences from nontarget organisms [20,21,22,23,24]. This approach has been widely used to suppress the amplification of 16S rDNA of host organelles (mitochondria and plastid) in plant and coral samples [22,23,24] and host 18S rDNA in mosquitoes and stool samples from mammals [20,21]. Locked nucleic acids (LNAs) are another type of artificial nucleic acid with a partially bridged structure, making it difficult to rotate, and stronger binding than DNA [25]. LNA-based blockers have been employed in PCR for denaturing gradient gel electrophoresis to inhibit the amplification of plant small subunit ribosomal RNA genes and to simultaneously detect plant-associated bacteriome [26]. Although both blockers, PNA and LNA, are expected to bind to the target sequences, their blocking efficiency may differ according to the experimental conditions. In fact, it is known that the binding capacity of DNA and LNA increases with high salt concentration, whereas PNA is not substantially affected by salt concentration [25,27]. It is unclear how this affects the strength and selectivity of amplification inhibition by artificial nucleic acids, and there are no directly relevant reports.

Another strategy to analyze the eukaryotic microbiome in host-associated samples is the use of primers specifically designed to target a particular group of organisms instead of universal primers. UNonMet-PCR, a method that selectively amplifies the 18S rDNA of non-metazoan organisms, was first reported for the detection of protists in oysters [28] and has since been used for the comprehensive detection of protists in ctenophores, corals, and human stool samples in combination with NGS [29,30]. As ticks are also metazoans, this method can theoretically be applied to investigate eukaryotes in ticks.

The aims of this study are to develop the methods for tick eukaryotic microbiome analysis by designing blocker PNA and LNA, which selectively bind to tick 18S rDNA and inhibit its amplification, and to compare the results between the blockers and the UNonMet-PCR method. Our results indicated that all the methods effectively suppressed the amplification of tick 18S rDNA and enabled the detection of diverse eukaryotes in ticks.

## 2. Materials and Methods

### 2.1. Ticks and DNA Extraction

A total of 17 adult tick DNA samples from three species, *Amblyomma testudinarium* (*n* = 5), *Haemaphysalis longicornis* (*n* = 6), and *Ixodes persulcatus* (*n* = 6), were employed. These samples were obtained from a previous study on rickettsial pathogens [31]. In brief, each tick was washed once with 1 mL of 10% sodium hypochlorite, twice with 1 mL of 70% ethanol, and once with 1 mL of sterile phosphate-buffered saline to decontaminate the tick body surface. The ticks were then crushed with 4.8 mm stainless steel beads (TOMY, Tokyo, Japan) using the Micro Smash MS-100R (TOMY) in 100 µL of Dulbecco’s Modified Eagle Medium (Gibco, Life Technologies, Gland Island, NY, USA). DNA was extracted from 50 µL of the tick homogenates using the blackPREP Tick DNA/RNA Kit (Analytik Jena, Jena, Germany), according to the manufacturer’s instructions. The details of tick samples (tick ID, tick species, sex, collection site, and year) are provided in Table 1.

### 2.2. Design of Blocker Nucleic Acids

We searched for the tick-specific sequences where the blocker nucleic acids can bind, which results in the inhibition of tick DNA amplification while allowing the amplification of other eukaryotic DNA by PCR. The sequences of the V4 hypervariable region of the 18S rDNA of Ixodidae (taxid: 6939), Ixodida (taxid: 6935), and Acari (taxid: 6933) were downloaded from NCBI GenBank (accessed on 26 November 2015). After manual quality filtering, the sequences were used to align the forward and reverse primer binding sites (TAReuk454FWD1 and TAReukREV3; Figure 1 and Table 2) using MEGA7 [32] and PROSEQ [33]. Based on the alignments, a median-joining network analysis (network analysis) was performed using Network ver. 4.6.1.6 [34] based on 41 bases in the region flanking each primer binding site. The blocker site was selected based on the following criteria: (1) no mismatches among tick sequences, (2) mismatches with apicomplexan parasite sequences, (3) T_m_ value of >70 °C, and (4) low self-complementarity. PNA (TickB_PNA) and LNA (TickB_LNA) were synthesized by PANAGENE (Daejeon, Korea) and GeneDesign Inc. (Osaka, Japan), respectively.

### 2.3. Plasmid Preparation

To evaluate the blocking efficiency of the designed blocker nucleic acids, the plasmids with the insertion of a partial 18S rDNA fragment of a tick (blocking target) and an Apicomplexan parasite (amplification target) were prepared. The 18S rDNA sequences of a tick *(H. longicornis* laboratory strain) and an Apicomplexan parasite (*Theileria orientalis* Chitose strain) were amplified using primer sets Cloning F and Cloning R (Table 2), designed against the regions >16 bp upstream or downstream of the universal primer binding sites (Table 2). Each PCR was performed using high-fidelity KOD-Plus-Neo DNA polymerase (Toyobo, Osaka, Japan) in a 25.0-µL reaction mixture containing 2.5 µL of 10× buffer for KOD-Plus-Neo, 300 nM of each primer, 2.5 µL of 2 mM dNTPs, 1.5 µL of 25 mM MgSO_4_, 0.5 units of KOD-Plus-Neo DNA polymerase, and 0.5 µL of template DNA; molecular-grade water was used to adjust the volume. The reaction conditions were as follows: 94 °C for 2 min and 45 cycles of 98 °C for 10 s, 55 °C for 30 s, and 68 °C for 30 s, and a final extension at 68 °C for 2 min. The PCR products were A-tailed using 10× Attachment Mix (Toyobo) and then cloned into a T-vector pMD20 (TaKaRa Bio Inc., Shiga, Japan). The vector was then transformed into ECOS^TM^ Competent *E. coli* DH5α (NIPPON GENE, Tokyo, Japan), and blue/white selection was conducted. The plasmid DNA was purified using NucleoSpin^®^ Plasmid EasyPure (Takara Bio Inc.). The concentrations of the plasmid DNA were measured using a Qubit dsDNA HS Assay Kit (Thermo Fisher Scientific, Waltham, MA, USA), and the corresponding copy numbers were calculated. The standard plasmids at the concentrations of 10^8^ copies/µL were obtained for a partial 18S rDNA fragment of *H. longicornis* (HlP) and *T. orientalis* (ToP).

### 2.4. PCR Amplification with the Addition of TickB_PNA and TickB_LNA

PCR was performed using the plasmids HlP and ToP as mock templates with the addition of artificial nucleic acids. Either of the blocker nucleic acids (TickB_PNA or TickB_LNA) was added to the PCR mixture before the PCR reaction started. Each PCR was performed in a 25.0 μL reaction mixture containing 12.5 μL of 2× KAPA HiFi HotStart ReadyMix (KAPA Biosystems, Wilmington, MA, USA), 200 nM of each primer (TAReuk454FWD1 and TAReuREV3), 2.5 μL of plasmid template (HlP 10^6^ copies/μL, ToP 10^6^ copies/μL, or ToP 10^2^ copies/μL), and 0, 1, 5, 10, 50, or 100 pmol TickB_PNA or TickB_LNA. The reaction conditions were set at 95 °C for 3 min, followed by 35 cycles of 98 °C for 30 s, 50 °C for 30 s, 72 °C for 30 s, and 72°C for 10 min. The copy numbers of the tick 18S rDNA in the DNA samples extracted with the blackPREP Tick DNA/RNA Kit ranged from 3.6 × 10^4^ /µL to 5.8 × 10^6^ /µL (*n* = 24; data not shown), and the plasmid copy number in this assay was determined. A total of 5.0 µL of each PCR product was mixed with 1.0 µL of loading dye (NIPPON GENE) and was loaded on 1.5% agarose S (NIPPON GENE) stained with Gel Red (Biotium, San Francisco, CA, USA) for electrophoresis at 100 V for 30 min. After separation, the gels were captured using an Image Quant LAS 4000 (GE Healthcare Life Science, Tokyo, Japan) at an exposure time of 1/30 s. The fluorescence intensity of each band was measured using 1D Gel Analysis of Image Quant TL (version 8.1; GE Healthcare Life Science). PCR and fluorescence intensity measurements were performed in triplicate.

### 2.5. UNonMet-PCR Amplification

UNonMet-PCR was performed in accordance with a method described in a previous study [30]. The first PCR was performed in a total volume of 25.0 µL, containing 12.5 µL of 2 × KAPA HiFi HotStart ReadyMix, 200 nM of each primer (18S-EUK581-F and 18S-EUK1134-R; Table 2), and 2.5 µL of template DNA (Table 1), and the volume was adjusted with molecular-grade water. The reaction was performed at 95 °C for 3 min, followed by 35 cycles of 95 °C for 30 s, 51.1 °C for 30 s, and 72 °C for 1 min, and a final extension at 72 °C for 5 min. The first PCR products were purified using the NucleoSpin Gel and PCR Clean-Up Kit (Takara Bio Inc.) and used as templates for the second PCR. The second PCR mixture was set as in the case of the first PCR, except the primers were replaced with EUK572F and 18S-EUK1009R, and 1.0 µL of the purified first PCR product was used as the DNA template. The reaction was performed at 95 °C for 3 min, followed by 25 cycles of 95 °C for 30 s, 55 °C for 30 s, and 72 °C for 1 min and a final extension at 72 °C for 5 min.

### 2.6. Illumina Library Preparation and MiSeq Run

To compare the blocking methods and the library purification methods, a total of 17 tick DNA samples listed in Table 1 were subjected to the conventional PCR without blocker (control), PCR with TickB_PNA or TickB_LNA, and UNonMet-PCR. Blockers (TickB_PNA or TickB_LNA) were added at a final concentration of 50 pmol/reaction, and PCR reactions were conducted in the same conditions described above. The PCR products were purified using the Agencourt AMPure XP Kit (AMPure; Beckman Coulter Inc., Brea, CA, USA) and subjected to index PCR using the Nextera XP Kit (Illumina, San Diego, CA, USA), according to the manufacturer’s protocol. Then, the index PCR products were purified using AMPure or NucleoMag NGS Clean-up and Size Select (SizeSelect; Macherey-Nagel, Düren, Germany). The former product is generally used for the purification of >100 bp DNA fragments, while the latter product can be used to enrich DNA fragments ranging in size from 150 to 800 bp. The purification protocol with AMPure was the same as that used for post-PCR purification. Fragment size was selected using SizeSelect according to the manufacturer’s protocol, with slight modifications. In brief, 40 µL of the index PCR product was mixed with 22 µL of NucleoMag NGS Bead Suspension. The supernatant obtained after separation with magnetic beads was transferred to the wells of a new plate, and the remaining beads were discarded to remove the larger PCR fragments. To remove the smaller PCR fragments, the supernatant was mixed with 8 µL of the bead suspension. The supernatant containing smaller fragments separated with magnetic beads was also discarded, and the PCR fragments attached to the magnetic beads were finally retrieved according to the manufacturer’s instructions. The length of the purified product was analyzed using the Agilent 2100 Bioanalyzer System (Agilent, Santa Clara, CA, USA). The libraries were then mixed at the same concentration, based on the concentration measured using a fluorescence-based method with a Qubit dsDNA BR Assay Kit (Thermo Fisher Scientific), and the pooled library was sequenced using an Illumina MiSeq platform and the MiSeq Reagent Kit v3 (600 cycles). Raw sequence data have been deposited in the DNA Data Bank of the Japan Sequence Read Archive under the accession number DRA011889.

### 2.7. Data Processing and Analysis

After demultiplexing and the merging of forward and reverse paired-end reads using QIIME2 (version 2019.10.0) [37], the DADA2 plugin in qiime2R [38] was used for quality filtering and the removal of chimeric sequences to produce a feature table of amplicon sequence variants (ASVs). Potential contaminants were identified using the “Decontam” R package with a threshold of 0.1 [39]. The contaminants were filtered out from all samples using QIIME2. Taxonomic assignments were made using the SILVA classifier (release 138), and ASVs identified as archaea, bacteria, or unidentified to the supergroup level were removed. To compare the performance of each PCR method, the proportion of non-tick reads (eukaryote reads excluding tick reads) was calculated, and pairwise comparisons between the PCR methods were performed using the pairwise.prop.test function with Bonferroni correction in R. In addition, three metrics of alpha diversity (Shannon diversity, Faith’s phylogenetic diversity (PD), and number of observed sequence variants (ASVs)) were calculated using QIIME2, exported to the qiime2R package, and visualized using the R packages ggplot2 and phyloseq [40]. To determine significant differences in alpha diversity among the PCR methods and the purification methods, we used the restricted maximum likelihood (REML) estimates of the parameters in linear mixed-effects models by using the lmer function in the R package lme4 [41]. The response variable was log-transformed alpha diversity, the collection site was the random variable, and the explanatory variables were the blocking methods, including PCR and purification methods. Principal coordinate analysis (PCoA) plots were generated to compare the protist communities between tick species based on Jaccard dissimilarity using QIIME2 and visualized as described above. Linear discriminant analysis effect size (LEfSe) was used to compare the relative abundances of different microbial genera among the control, blockers, and UNonMet-PCR [42].

## 3. Results

### 3.1. Designing of TickB_PNA and TickB_LNA

The target sequences for the TAReuk454FWD1 primer were obtained from ixodid ticks (*n* = 250), argasid ticks (*n* = 167), and mites (*n* = 23). A network analysis of the region flanking the primer showed that ixodid ticks could be split into several haplotypes, and a conserved region among the tick sequences was not identified (Appendix A). The sequences in the TAReukREV3 primer region were found in ixodid ticks (*n* = 111), argasid ticks (*n* = 76), and mites (*n* = 97). Network analysis revealed that the ixodid ticks were clustered into two haplotypes (Appendix A). Furthermore, a sequence comparison of the region between 27 bp upstream and 1 bp downstream of the 3′ end of the reverse primer in ticks and almost all known apicomplexan parasites was performed. Finally, the region between 11 bp upstream and 6 bp downstream of the 3′ end of the TAReukREV3 primer, including 3 bp mismatches between ticks and apicomplexan parasites, was selected as the blocking site (Figure 1), and two different artificial nucleic acids were synthesized as follows: tick blocker PNA (TickB_PNA), 5′-GAT*C*AAWGAAAACATT*-3′ (* indicates nucleotide mismatch), tick blocker LNA (TickB_LNA), 5′- GAT*C*AAWGAAAACATT*-3′ (* indicates nucleotide mismatch; underline indicates LNA replacement).

### 3.2. Blocking Efficacy of TickB_PNA and TickB_LNA

Using HlP as a template at a final concentration of 2.5 × 10^6^ copies per reaction, the relative fluorescence intensity of the PCR products did not change significantly when 1 or 5 pmol of TickB_PNA was added, but it decreased in a dose-dependent manner when 10, 50, or 100 pmol TickB_PNA was added (Figure 2a). The relative fluorescence intensity of the PCR products increased in the presence of 1 or 5 pmol TickB_LNA but decreased when 50 or 100 pmol was added (Figure 2a). When ToP was used as a template at a final concentration of 2.5 × 10^6^ copies per reaction, the relative fluorescence intensity of each PCR product for both TickB_PNA and TickB_LNA showed an increasing pattern (Figure 2b). With ToP at a final concentration of 2.5 × 10^2^ copies per reaction as a template, the relative fluorescence intensity of the amplified product decreased according to the amount of TickB_PNA added to the reaction solution (Figure 2c). In contrast, there was no clear change in the relative intensity of the amplified product upon the addition of TickB_LNA to the reaction (Figure 2c).

### 3.3. Comparison of Purification Methods by a Fragment Length Analysis

Fragment length analysis of selected purified amplified products was performed using the Agilent 2100 Bioanalyzer System. When the PCR products without blockers were purified with AMPure, the fragment lengths of the purified products showed a single peak (Figure 3a). However, the PCR products with blockers contained multiple long fragment peaks in addition to the target fragment peak even after purification with AMPure (Figure 3b,d). In contrast, when the PCR products obtained with blockers were purified and size-selected by SizeSelect, a relatively low frequency of long fragments other than the major peak was observed (Figure 3c,e), although there was variation among samples (Appendix A).

### 3.4. Comparison of Blocking Efficacy of TickB_PNA, TickB_LNA, and UNonMet-PCR

Seventeen tick DNA samples were subjected to PCR without a blocker (control), PCR with TickB_PNA or TickB_LNA, and UNonMet-PCR. For PCRs with blockers, two purification methods (AMPure and SizeSelect) were employed. The purified amplicons were finally sequenced on a MiSeq platform. After quality filtering, joining steps, and decontamination, 6,724,825 reads were obtained and clustered into 18,901 ASVs (Appendix A). The remaining 2510 ASVs, including 6,163,967 reads (average 385,248 per sample, min 13, max 253,437), were subjected to further analysis after filtration of reads belonging to archaea and bacteria and unclassified ASVs at the kingdom level.

The proportions of non-tick reads to all eukaryote reads were significantly higher for PCR with blockers and UNonMet-PCR than those for control (*p* < 0.01), and the proportions of tick reads were lower (Figure 4 and Appendix A). The median proportion of non-tick reads to all eukaryote reads was 0.03% for the control method, while higher median values, 3.24%, 5.75%, 1.53%, 3.18%, and 1.37%, were obtained for TickB_PNA with AMPure, TickB_PNA with SizeSelect, TickB_LNA with AMPure, TickB_LNA with Size Select, and UNonMet-PCR, respectively (Appendix A). Tick ID 467 showed a high proportion (81.14%) of non-tick reads even with the control method. Figure 4 indicates the non-tick read enrichment value, calculated by dividing the proportion of non-tick reads of each method by that obtained for each individual control. The median non-tick read enrichment values for TickB_PNA with AMPure, TickB_PNA with SizeSelect, TickB_LNA with AMPure, TickB_LNA with SizeSelect, and UNonMet-PCR were 60.3, 103.8, 19.6, 32.9, and 35.2 folds, respectively (Appendix A). Upon using PNA-based methods, all samples except for tick ID 467 showed more than 10-fold enrichment (ranging between 12.8 and 705.0 folds). The decreased proportion of non-tick reads compared to control was only observed in tick IDs 66 and 2876 upon using UNonMet-PCR.

The linear mixed models showed significant differences in alpha diversity metrics among the PCR and purification methods (Figure 5, Appendix A). Upon comparing the control and each PCR method, we found that Shannon diversity and the number of observed ASVs were significantly higher for all of the newly established methods than for the control (*p* < 0.01) except for TickB_LNA with AMPure (*p* > 0.05). Faith’s PD increased significantly when blockers were added to the reaction mixture (*p* > 0.01), but not for UNonMet-PCR (*p* > 0.05). Upon comparing the blockers with UNonMet-PCR, we found that Shannon diversity was significantly higher for UNonMet-PCR (*p* < 0.01), whereas Faith’s PD was significantly higher when blockers were used (*p* < 0.01). In addition, the number of observed ASVs did not differ significantly between the blockers and UNonMet-PCR (*p* > 0.05). Furthermore, analysis of alpha diversity between the AMPure and SizeSelect purification methods for TickB_PNA and TickB_LNA, respectively, did not reveal significant differences among the three metrics (*p* > 0.05).

### 3.5. PCR with Blockers or UNonMet-PCR

The top 10 taxa detected in analyses using PCRs with TickB_PNA or TickB_LNA and UNonMet-PCR are shown in Table 3. Relative abundances of detected taxa in individual ticks are also shown in Appendix A. At a higher taxonomic level, fungi accounted for the highest proportion of reads for all methods. At the genus level, the proportion of *Capsicum, Papiliotrema,* and *Ascochyta* was the highest in TickB_PNA, TickB_LNA, and UNonMet-PCR, respectively. The Apicomplexa detected are listed in Table 4. Of note, a direct comparison of sensitivity to Apicomplexa between each method is not possible owing to the different depths that are dependent on the final number of reads obtained (Appendix A). The genus *Gregarina* was detected in 3 out of the 17 tick DNA samples, while the genera *Amoebogregarina, Cryptosporidium, Colpodellidae, Theileria*, Apicomplexa (unknown), and Eugregarinorida (the order to which *Gregarina* belongs) were detected in 1 out of the 17 tick DNA samples.

The relative abundances of the detected taxa were compared using LEfSe (all alpha values <0.05, logarithmic LDA score threshold = 2.0) among conventional PCR (control), PCRs with blockers, and UNonMet-PCR (Figure 6). In a comparison of the relative abundance between the control and each blocker by LEfSe, Arachnida, Holozoa, Animalia, and unclassified eukaryotes, including mites, were significantly more abundant in the control, whereas Nucletmycea, Arachnida, Fungi, Archaeplastida, Chloroplastida, Alveolata, Rhizaria, Ciliophora, and SAR supergroup (Stramenopiles, Alveolata and Rhizaria) were significantly more abundant when using the blocker methods (Figure 6a,b). Fungi and Alveolata were significantly more abundant for the UNonMet-PCR method than in the control; however, other taxa such as Archaeplastida, which showed elevated abundances when blockers were added, did not differ significantly between the UNonMet-PCR method and the control (Figure 6c). In addition, when UNonMet-PCR and blocker methods were compared, UNonMet-PCR detected significantly more fungi, whereas TickB_PNA and TickB_LNA detected more Alveolata, Chloroplastida, and unclassified eukaryotes (Figure 6e,f, Appendix A). In the comparison of TickB_PNA and TickB_LNA, unclassified eukaryotes or Opisthokonta were significantly more abundant in TickB_PNA (Figure 6d).

### 3.6. Comparison of Eukaryotic Microbiomes among Three Tick Species

PCoA of the eukaryotic microbiome detected using the TickB_PNA, TickB_LNA, and UNonMet-PCR methods was performed based on the Jaccard diversity index (Figure 7). The detection rates using TickB_PNA and TickB_LNA were 5.25% (median: 4.85%, min: 0.95%, max: 13.27%) and 2.51% (median: 1.79%, min: 0.02%, max: 9.01%), respectively, of “unclassified eukaryotes” on average (Appendix A). We selected the Jaccard diversity method, which does not use phylogenetic branch lengths and assigns more weight to rare species. As a result, the eukaryotic composition based on both blocker-based methods was somewhat separated depending on the tick species, with some overlap (for instance, *A. testudinarium* and *I. persulcatus* in TickB_LNA; Figure 7a,b). For the UNonMet-PCR method, the composition of the eukaryotic community in each sample was generally divided by tick species (Figure 7c).

## 4. Discussion

An increasing number of tick microbiome studies have provided fundamental data related to pathogen transmission to humans and animals and the complex interactions between microorganisms at the molecular level that are important for tick physiology. However, most of these studies have focused on bacterial and viral communities in ticks [18,43,44,45,46,47]. Considering that the arthropod microbiome involves complex interactions among prokaryotes, viruses, and eukaryotic microorganisms [6,9,48,49,50,51], the lack of data for eukaryotes may result in incomplete or misleading analyses. The aim of this study was to develop methods for the analysis of eukaryotes in ticks by selectively suppressing the amplification of tick 18S rDNA but allowing the amplification of sequences of other eukaryotes. To the best of our knowledge, this is the first comprehensive analysis of eukaryotes harbored by ticks using NGS.

The newly developed methods make it possible to simultaneously investigate the presence of known pathogenic protozoa, such as Apicomplexa parasites, as well as the diversity of microeukaryotes harbored by ticks (Figure 7 and Appendix A, Table 3 and Table 4). In comparison with the control method, in which almost all reads were derived from ticks, all newly established methods showed a higher frequency of non-tick reads, resulting in higher alpha diversity (Figure 4 and Figure 5 and Appendix A). In particular, PNA-based amplicon blocking with SizeSelect and UNonMet-PCR were effective for detection. It is worth mentioning that one sample (tick ID 467) showed an extremely high proportion (81.14%) of non-tick reads with the control method (Appendix A). Since all the sample processing procedures, including DNA extraction, are the same, one possible reason is the presence of a high abundance of fungi in the tested tick due to environmental contamination or infection to the tick. Another possible explanation is the presence of nucleotide mismatches at the primer binding sites of the 18S rDNA sequences of the tested tick, resulting in poor amplification of tick DNA by PCR. Nonetheless, the fact that the proportion of non-tick reads was greatly increased in all other samples by the blockers and UNonMet-PCR indicates that these techniques are useful for analyses of the eukaryotic microbiome in ticks. In addition, the TickB_PNA method used in our study for hard ticks can also be applied to analyses of the eukaryotes in some soft ticks and mites that share sequence similarity with the blockers designed in this study (Appendix A).

The TickB_PNA, TickB_LNA, and UNonMet-PCR strategies all increased Shannon diversity and observed ASVs compared with levels in the controls with only universal primers, proving that they can detect a greater diversity of microorganisms (Figure 5a,c). The use of blockers not only increased alpha diversity but also increased the frequency of eukaryotes compared with those obtained using the control and UNonMet-PCR methods (Figure 4 and Appendix A). This suggests that blockers are appropriate for targeting the entire eukaryotic microbiome of ticks. UNonMet-PCR resulted in a low Faith’s PD, and this value was not significantly different from that of the control, even though Shannon diversity and observed ASVs were comparable to those obtained with the blockers (Figure 6). Shannon diversity is calculated based on the proportion of species in the total sample, while Faith’s PD accounts for the length of the phylogenetic branches. Taken together, the proportion of matching sequences in eukaryotic communities in ticks was lower for UNonMet-PCR-primers than for the TAReuk primer pair, and, therefore, the phylogeny of sequences detected was weighted to fungi and Apicomplexa (Figure 5) [52]. UNonMet-PCR detects a significantly higher abundance of fungal-derived sequences than the blocker methods and is more suitable for specifically targeting these sequences (Figure 6e,f). There are some reports of entomopathogenic fungi in ticks and insects [14,53,54], and some fungal species have a symbiotic relationship with insects [10]. Therefore, an exhaustive search using UNonMet-PCR of fungi harbored by ticks and their effects would be useful for establishing novel methods for tick control.

A comparative analysis showed that TickB_PNA may be superior to TickB_LNA based on the proportion of non-tick reads (Figure 4 and Appendix A) and alpha diversity (Figure 5). When plasmids were used as PCR templates, TickB_PNA suppressed the amplification of tick DNA more strongly than TickB_LNA at the same concentration (Figure 2). In the amplicon analysis, TickB_PNA showed significantly higher alpha diversity and a higher number of eukaryotic reads than those of TickB_LNA (Figure 6), in accordance with the results obtained from plasmid templates. These results suggest that the TickB_PNA-based PCR method is more effective for detecting eukaryotes than the TickB_LNA-based method used in this study. However, when ToP was used as a template at 2.5 × 10^2^ copies/reaction (a nontarget for amplification inhibition), TickB_PNA inhibited amplification with increasing addition, whereas less notable inhibition of amplification was observed with TickB_LNA. In addition, only 11 of the 16 bases of TickB_LNA were synthesized with LNAs because the aim of the study was to compare these with PNA at the same T_m_. Therefore, the inhibition of tick DNA amplification is expected to increase by replacing more bases or all bases with LNAs, as this will improve the ability to bind to complementary DNA. Considering these points, we cannot definitively conclude that PNA is more suitable than LNA for the amplification and inhibition of large amounts of DNA in microbial analyses by NGS. Although many examples of the application of blocker PNAs to amplicon preparation have been reported [20,21,22,23], blocker LNAs have not been used for comprehensive analyses by NGS. It is clear that selective amplification inhibition using PNA is useful for amplicon analyses [20,21,22,23,24]; however, for the reasons mentioned above, LNA may also be appropriate depending on the conditions, including the blocker sequences and buffers, and more trials are needed to optimize the choice of artificial nucleic acids for blocking. 

In this study, the genus *Gregarina* (phylum Apicomplexa) was detected in 18% (3/17) of the tick samples (Appendix A and Table 4). This is the first report of its detection in ticks, while *Gregarina* has been detected in a wide variety of metazoans, especially in arthropods [55]. In addition, although many other apicomplexan protozoans are parasitic, such as *Babesia, Theileria,* and *Cryptosporidium* [56], *Gregarina* are highly diverse, ranging from parasitic to symbiotic [55]. There are reports of symbiotic *Gregarina* in arachnids [57], as well as in blood-sucking insects such as sand flies and mosquitoes [58,59,60]. Further studies are needed to determine the effects of tick-borne *Gregarina* on hosts, with the aim of devising new tick-control strategies.

At higher taxonomic levels, fungi, Charophyta (freshwater grasses), Ciliophora, and Cercozoa were common. At the genus level, *Capsicum* (Eggplant, Capsicum), *Mykophagophrys* (phylum Ciliophora), *Ascochyta* (phylum Ascomycota), *Cladosporium* (phylum Ascomycota), *Aspergillus* (phylum Ascomycota), and *Didymella* (phylum Ascomycota) were commonly observed (Table 3). Some species of fungi are pathogenic or symbiotic in arthropods [10,54], and their spores are scattered in the environment. The spores of Fungi, Ciliophora [61], and Cercozoa [62,63] are ubiquitous in soil and freshwater. The reason for the frequent occurrence of *Capsicum* is not known, but Charophyta and *Capsicum* are likely to occur in the environment. Although the possibility of contamination from the body surface of ticks with these microbial DNA cannot be completely excluded, the body surfaces were thoroughly cleaned by washing with 10% sodium hypochlorite and 70% ethanol in the preliminary stages of DNA extraction [64]; it is likely that the origin of the detected protists was mainly internal. Ticks obtain water transdermally from water vapor when not attached to a host [65,66]; however, *Amblyomma americanum* feeds not only from water vapor but also from liquid water and takes up water in its salivary glands and intestinal tract [67]. The frequent detections of taxa that appear to be from the environment may indicate that other ticks also consume liquid water. The potential for the uptake of microorganisms from the environment is an important consideration, as it may disrupt the tick microbiome.

In the PCoA plot, specimens were divided according to species when using blockers and were clearly divided by species when applying UNonMet-PCR (Figure 7). These results suggest that tick species each possess a unique eukaryotic microbiota. A certain degree of overlap between species was observed for all methods, and this can be explained by various factors such as the unknown effects of unclassified eukaryotes and environmental factors that are known to affect the bacterial community [16,68]. Furthermore, bacterial flora tends to be similar in different tick species, and we detected a similar tendency for eukaryotes, suggesting that ticks may provide an environment in which certain microorganisms can be readily taken up or that they harbor organisms that have evolved together.

When the blocker was added to the PCR, the nonspecific amplification of the long-chain increased (Figure 3). Based on this nonspecific amplification without an adaptor sequence, we expected quantitative real-time PCR (qPCR) to accurately measure the concentration; however, the estimated concentration was the same as that of Qubit measurement (data not shown). Thus, we determined that the adapter sequence was attached to the long-chain sequence and accurate measurements were impossible, even by qPCR. These nontarget sequences should be excluded because they negatively affect concentration adjustment and cluster formation and reduce the final number of acquired reads, even though the origin of these remains unknown. In the present study, the nonspecific amplification was successfully mitigated using SizeSelect instead of AMPure, and the diversity of the detected sequences also increased (Figure 5). Unfortunately, it is not clear from this study whether the presence of nonspecific amplification is specific to our samples or not. Therefore, it is recommended to evaluate the library size first when using the TickB_PNA and TickB_LNA methods developed in this study.

When an amplicon analysis was performed using blockers, a large proportion of unclassified eukaryote sequences in existing databases was observed (Figure 6, Appendix A). These unknown sequences have also been found in 18S rDNA amplicon analyses using other blocker PNAs [21,69], suggesting that these sequences are a mixture of previously unknown eukaryotic rDNA and nonspecific products outside the target region. Protists are not only present in the microbial community carried by metazoans [55,69] but are also becoming more important components of the environment, as indicated by an increasing number of NGS analyses [52,69,70,71]. Therefore, as the number of registered sequences increases in the future, the issue of unclassified sequences in high-throughput analyses will be resolved.

We have demonstrated a comprehensive analysis of eukaryotic organisms in ticks using the blocker and UNonMet-PCR methods for high-throughput sequencing. In addition to known tick-borne protozoa, a wide variety of previously unreported protists were detected. We recommend the use of blocker methods to capture the entire eukaryotic microbiome of ticks. The UNonMet-PCR method is also useful in the characterization of the fungal community in ticks. In the future, these efforts will help uncover the eukaryotic microbiome in ticks, which may also contribute to the development of novel control strategies for ticks and vector-borne pathogens.

## Figures and Tables

**Figure 1 microorganisms-09-01051-f001:**
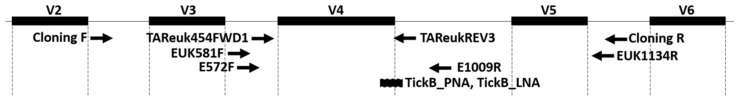
Locations of the primers/blockers used in this study. The upper horizontal bars indicate the hypervariable regions (V2, V3, V4, V5, and V6) of 18S rRNA gene. Each arrow indicates the position and orientation of the primer, while a wavy line represents the position of the blockers. Primer/blocker names are provided next to the arrows or the wavy line.

**Figure 2 microorganisms-09-01051-f002:**
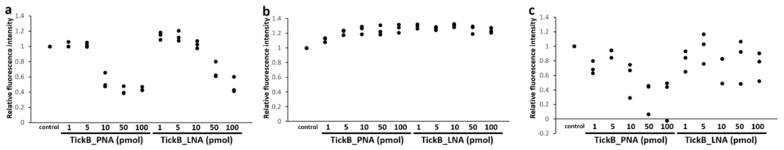
Change in the relative fluorescence intensity of PCR products in the presence of various concentrations of blocker nucleic acids. In each graph, the *x*-axis shows the type and amount of blocker (TickB_PNA or TickB_LNA) added to the reaction mixture, and the *y*-axis shows the relative fluorescence intensity of the bands. PCR and fluorescence intensity measurements were performed in triplicate. For each trial, the fluorescence intensity in the absence of the blocker (control) was set to 1.0, and the relative fluorescence intensity with the blocker is presented as a relative value. The types and concentrations of plasmids as templates were (**a**) partial *Haemaphysalis longicornis* 18S rRNA gene at 2.5 × 10^6^ copies/reaction, (**b**) partial *Theileria orientalis* 18S rRNA gene (ToP) at 2.5 × 10^6^ copies/reaction, or (**c**) ToP at 2.5 × 10^2^ copies/reaction.

**Figure 3 microorganisms-09-01051-f003:**
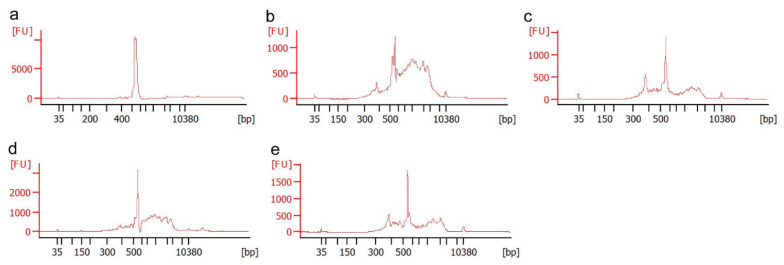
Fragment size distribution of amplicon libraries using different PCR and purification methods. *Haemaphysalis longicornis* (tick ID 3631) DNA was used as a template. The *X*- and *y*-axes show detected fragment size and fluorescent units, respectively. (**a**) PCR without blocker and AMPure purification; (**b**) PCR with TickB_PNA and AMPure purification; (**c**) PCR with TickB_PNA and SizeSelect purification; (**d**) PCR with TickB_LNA and AMPure purification; (**e**) PCR with TickB_LNA and SizeSelect purification.

**Figure 4 microorganisms-09-01051-f004:**
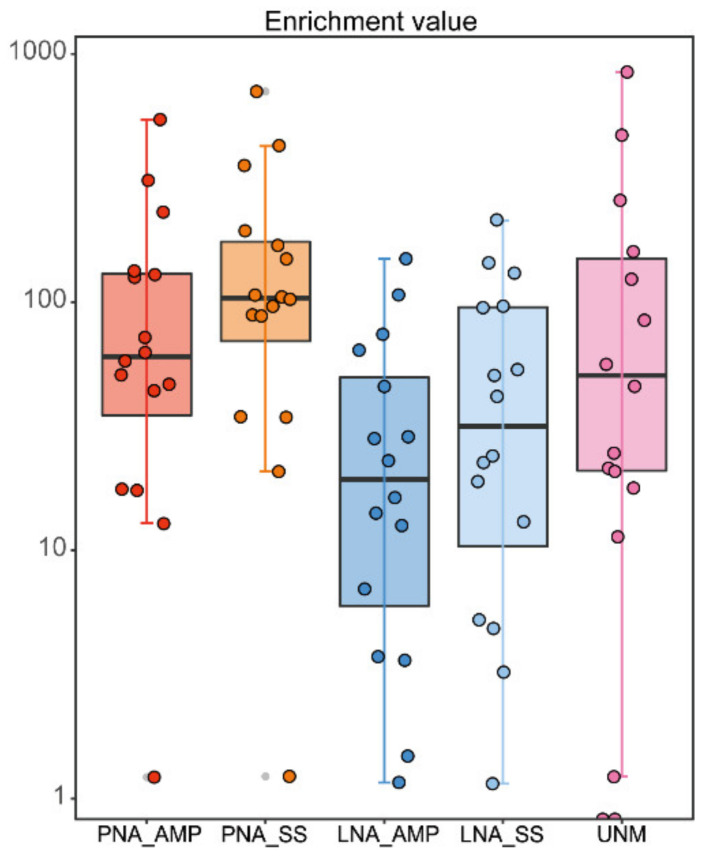
Comparison of non-tick read enrichment values calculated for each method. The non-tick read enrichment value was calculated by dividing the proportion of non-tick reads of each method by that obtained for each individual control. Universal primer sets TAReuk454FWD1 and TAReukREV3 were used for PCRs of the control, TickB_PNA (PNA), and TickB_LNA (LNA). The primer sets EUK581F and EUK1134R were used for the first PCR, and E572F and E1009R were used for the second PCR for UNonMet-PCR (UNM). The purification methods were AMPure (AMP) or SizeSelect (SS). Tick ID 3611 was filtered out because only tick reads were detected in the control method. The non-tick read enrichment value is shown on as a logarithmic scale. Outlier plots are indicated by additional dots in grey.

**Figure 5 microorganisms-09-01051-f005:**
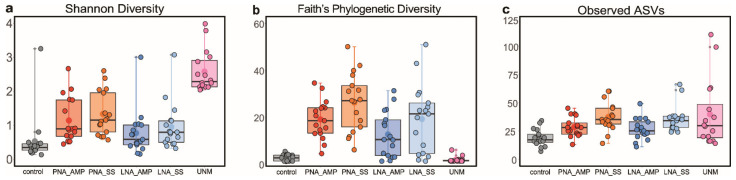
Comparison of the efficiency of blocking methods based on alpha diversity. Universal primer sets TAReuk454FWD1 and TAReukREV3 were used for PCR of the control, TickB_PNA (PNA), and TickB_LNA (LNA). The primer sets EUK581F and EUK1134R were used for the first PCR, and E572F and E1009R were used for the second PCR for UNonMet-PCR(UNM). The purification methods were AMPure (AMP) or SizeSelect (SS). The diversity indexes used were (**a**) Shannon diversity, (**b**) Faith’s phylogenetic diversity, and (**c**) the number of observed amplicon sequence variants (ASVs). Outlier plots are indicated by additional dots in grey.

**Figure 6 microorganisms-09-01051-f006:**
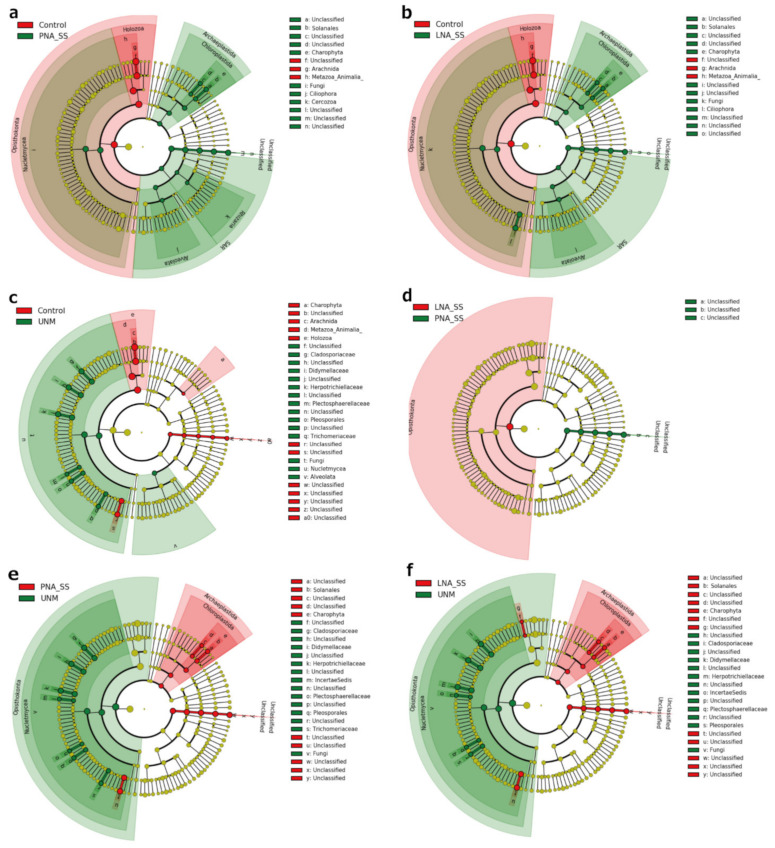
Cladogram showing abundant eukaryotic taxa detected upon using the control, TickB_PNA, TickB_LNA, and UNonMet-PCR methods. SizeSelect was used as a purification method for all blocker-based amplification products. The universal primer sets TAReuk454FWD1 and TAReukREV3 were used for control PCR, TickB_PNA, and TickB_LNA. The primer sets EUK581F and EUK1134R for the first PCR and E572F and E1009R for the second PCR were used for UNonMet-PCR (UNM). LEfSe was conducted for comparisons between (**a**) the control and TickB_PNA, (**b**) the control and TickB_LNA, (**c**) the control and UNM, (**d**) TickB_PNA and TickB_LNA, (**e**) TickB_PNA and UNM, and (**f**) TickB_LNA and UNM. LEfSe analysis was conducted, including tick reads.

**Figure 7 microorganisms-09-01051-f007:**
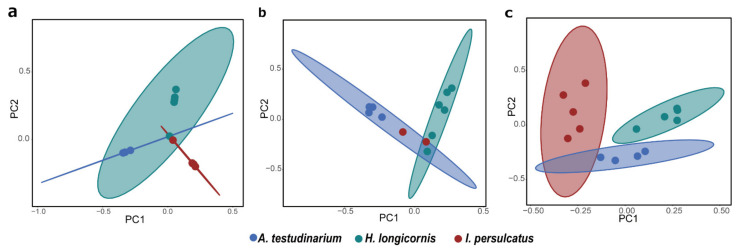
Principle coordinate analysis (PCoA) of eukaryotic microbiome composition in three tick species, *Amblyomma testudinarium, Haemaphysalis longicornis,* and *Ixodes persulcatus.* Arachnida (ticks) amplicon sequence variants (ASVs) were excluded, and the Jaccard diversity was calculated based on the microeukaryotic sequences using the R package phyloseq. Amplicons obtained using the TickB_PNA- and TickB_LNA-based methods (see (**a**,**b**)) were purified with SizeSelect. For TickB_PNA, 16 samples were included, and only tick ID 3611 was filtered out owing to the low number of sequences after filtration. For TickB_LNA, 13 samples were included, and tick IDs 66, 210, 258, and 933 were filtered out owing to the low number of remaining sequences after filtration. For UNonMet-PCR (**c**), only 15 samples were included because two samples (tick IDs 66 and 2876) with 100% Acari were filtered out. Points representing the same tick species are indicated by circles.

**Table 1 microorganisms-09-01051-t001:** Tick samples used in this study.

Tick ID	Tick Species	Sex	Prefecture	Location	Year
**66**	*I. persulcatus*	F	Hokkaido	42.61, 141.95	2013
**210**	*I. persulcatus*	F	Hokkaido	42.98, 142.80	2013
**258**	*I. persulcatus*	F	Hokkaido	43.08, 142.01	2013
**467**	*I. persulcatus*	M	Hokkaido	43.36, 142.61	2013
**932**	*I. persulcatus*	M	Hokkaido	43.92, 142.86	2014
**933**	*I. persulcatus*	M	Hokkaido	43.92, 142.86	2014
**2592**	*A. testudinarium*	M	Shimane	35.04, 132.68	2016
**2874**	*A. testudinarium*	F	Ehime	33.35, 132.02	2017
**2876**	*A. testudinarium*	F	Kochi	33.27, 132.99	2017
**3148**	*A. testudinarium*	F	Ehime	33.63, 132.56	2017
**3149**	*A. testudinarium*	F	Ehime	33.63, 132.56	2017
**3332**	*H. longicornis*	F	Hiroshima	34.49, 132.45	2017
**3611**	*H. longicornis*	F	Chiba	35.19, 140.25	2018
**3620**	*H. longicornis*	M	Chiba	35.19, 140.25	2018
**3631**	*H. longicornis*	M	Wakayama	33.83, 135.30	2018
**3643**	*H. longicornis*	F	Wakayama	34.37, 135.64	2018
**3648**	*H. longicornis*	M	Kochi	33.29, 134.17	2018

F, female; M, male.

**Table 2 microorganisms-09-01051-t002:** Primers and blockers used in this study.

Name	Sequences (5′→3′)	Application	References
**Cloning F**	CACATCTAAGGAAGGCAGCA	Plasmid preparation	This study
**Cloning R**	CCCTTCCGTCAATTCCTTTA		This study
**M13 Primer M4**	GTTTTCCCAGTCACGAC	Plasmid insert sequence identification	Takara Bio Inc.
**M13 Primer RV**	CAGGAAACAGCTATGAC		
**TAReuk454FWD1**	CCAGCASCYGCGGTAATTCC	Universal primer sets amplifying V4 region of 18S rDNA	[35]
**TAReukREV3**	ACTTTCGTTCTTGATYRA		
**EUK581F**	GTGCCAGCAGCCGCG	UNonMet-PCR (first PCR)	[30]
**EUK1134R**	TTTAAGTTTCAGCCTTGCG		
**E572F**	CCATCTCATCCCTGCGTGTCTCCGACTCAG	UNonMet-PCR (second PCR)	[36]
**E1009R**	CCTATCCCCTGTGTGCCTTGGCAGTCTCAG		
**TickB_PNA**	GATCAAWGAAAACATT	Blocking the amplification of tick 18S rDNA	This study
**TickB_LNA**	GATCAAWGAAAACATT ^1^	Blocking the amplification of tick 18S rDNA	This study

^1^ Underlined sequences are LNA replacements.

**Table 3 microorganisms-09-01051-t003:** Top 10 observed taxa based on the TickB_PNA, TickB_LNA, or UnonMet-PCR methods at a higher taxonomic level and the genus level.

Rank	TickB_PNA-SizeSelect		TickB_LNA-SizeSelect		UNonMet-PCR	
	Higher Category	Genus Level	Higher Category	Genus Level	Higher Category	Genus Level
**1**	Fungi (0.8649%)	*Capsicum* (0.0703%)	Fungi (1.3424%)	*Papiliotrema* (0.1527%)	Fungi (6.9371%)	*Ascochyta* (1.1681%)
**2**	Charophyta (0.1386%)	*Gregarina* (0.0396%)	Charophyta (0.0913%)	*Cladosporium* (0.0963%)	Ciliophora (0.1170%)	*Cladosporium* (1.0394%)
**3**	Ciliophora (0.0715%)	*Mykophagophrys* (0.0306%)	Ciliophora (0.0675%)	*Ascochyta* (0.0913%)	Cercozoa (0.0612%)	*Aspergillus* (0.7208%)
**4**	Apicomplexa (0.0614%)	*Ascochyta* (0.0230%)	Apicomplexa (0.0133%)	*Capsicum* (0.0484%)	Apicomplexa (0.0502%)	*Didymella* (0.2340%)
**5**	Cercozoa (0.0310%)	*Cladosporium* (0.0163%)	Cercozoa (0.0117%)	*Vishniacozyma* (0.0336%)	Flabellinia (0.0113%)	*Penicillium* (0.0831%)
**6**	Retaria (0.0083%)	*Heteromita* (0.0135%)	Ochrophyta (0.0049%)	*Colpoda* (0.0256%)	Dinoflagellata (0.0065%)	*Didymosphaeriaceae* (0.0686%)
**7**	-	*Colpoda* (0.0127%)	Retaria (0.0020%)	*Mykophagophrys* (0.0203%)	Cryptomonas (0.0048%)	*Roussoella* (0.0503%)
**8**	-	*Spencermartinsiella* (0.0048%)	Chlorophyta (0.0019%)	*Gregarina* (0.0097%)	Didymium (0.0036%)	*Gregarina* (0.0357%)
**9**	-	*Meira* (0.0048%)	Dinoflagellata (0.0004%)	*Heteromita* (0.0087%)	Euamoebida (0.0013%)	*Heteromita* (0.0293%)
**10**	-	*Aspergillus* (0.0043%)	-	*Piskurozyma* (0.0084%)	Protostelium sp. (0.0012%)	*Sorogena* (0.0284%)

Percentages in brackets indicate the abundance of reads for each phylum relative to the total eukaryotic reads for each PCR method. Acari (ticks) and unclassified eukaryotes were excluded from the ranking.

**Table 4 microorganisms-09-01051-t004:** Detected genera in the phylum Apicomplexa.

Tick ID	Control	TickB_PNA-AMPure	TickB_PNA-SizeSelect	TickB_LNA-AMPure	TickB_LNA-SizeSelect	UNonMet-PCR
**66**	-	-	-	-	-	-
**210**	-	-	-	-	-	Cr
**258**	-	-	-	-	-	-
**467**	-	-	-	-	-	T
**932**	-	-	-	-	-	-
**933**	-	-	-	-	-	-
**2592**	-	G, A	-	A	-	G, A
**2874**	N	G, N	G, N	G, N	G, N	G, EU, N, Co
**2876**	-	-	-	-	-	-
**3148**	-	-	-	-	-	-
**3149**	-	-	-	-	-	-
**3332**	-	G	G	G	G	G
**3611**	-	-	-	-	-	-
**3620**	-	-	-	-	-	-
**3631**	-	-	-	-	-	-
**3643**	-	-	-	-	-	-
**3648**	-	-	-	-	-	-

N, unclassified (phylum) Apicomplexa; A, *Amoebogregarina* (order Eugregarinorida); G, *Gregarina* (order Eugregarinorida); Cr, *Cryptosporidium* (order Euoccidiorida); EU, unclassified order Eugregarinorida; Co, unclassified family Colpodellidae (order Colpodellida).

## Data Availability

Raw sequence data have been deposited in the DNA Data Bank of Japan (DDBJ) Sequence Read Archive under accession number DRA011889.

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
