# Peer review of "Applications of Blocker Nucleic Acids and Non-Metazoan PCR Improves the Discovery of the Eukaryotic Microbiome in Ticks"

_microorganisms, 2021, doi:10.3390/microorganisms9051051_

Round 1

Reviewer 1 Report

This is an important and timely study that provides an analysis of several available valuable tools for better querying the eukaryotic component of tick microbiomes. As the authors clearly articulate, the tick 18S rDNA inhibits successful amplification of eukaryotic microorganisms in ticks. The authors examined three approaches to block the host DNA are presented, PNA, LNA, and UNonMet-PCR and also evaluated a couple of different size selection methods. The introduction could benefit from a better description of the three approaches and the two clean-up methods: specifically, what is the general approach, how they are different, advantages, and disadvantages?

I would also like to see more biological interpretation of the different results as measured by Shannon’s vs. Faith’s diversity indices. What is the significance of different responses of the blocking methods in these two types of methods?

The other area that could benefit from additional clarification is in the methods. Given that many of these approaches are relatively novel and not widely used, it would be helpful for the methods to start with a general introduction to what the goal of each step is. For instance, in section 2.2 Designing Blocker Nucleic Acids, a broad overview of how the blocker works would help explain the design details. Does the blocker nucleic acid get added to the sample before the PCR? During? And in section 2.3 Plasmid preparation, what is the purpose of generating plasmids of tick and apicomplexan 18S DNA?

The discussion is quite thorough but could use some more definitive assessments of the usefulness of the methods tested here. It appears that PNA is effective for incrasing species richness, but then there are some cases where LNA might be better. UnonMet may also be better in terms of identifying more taxa in more samples. What are the recommendations for each of these methods with respect to different kinds of research questions or objectives. How should researchers decide which of these to use?

Also, what is the significance of the SizeSelect method working better than Ampure? How are they different? Is this a general pattern or is it because of the particular amplicons in these samples?

Line 88: needs to be edited for grammar. Suggest: “Herein, we report on the use of new methods…” Although are these new? Several citations are provided that have used the PNA and LNA methods go back to 2012 although several are more recent. Perhaps this whole last paragraph of the introduction could be rephrased to summarize the main goal of the study, to evaluate these three approaches in a tick system.

Line 176-177: What does it mean that PCR with TickB_PNA or TickB-LNAa… were added with a final concentration? Were these added to the conventional PCR? The blocked PCRS were added to a conventional PCR?  A clearer description of when LNA and PNA blocking occurs would really be helpful.

Author Response

Reviewer 1:

This is an important and timely study that provides an analysis of several available valuable tools for better querying the eukaryotic component of tick microbiomes. As the authors clearly articulate, the tick 18S rDNA inhibits successful amplification of eukaryotic microorganisms in ticks. The authors examined three approaches to block the host DNA are presented, PNA, LNA, and UNonMet-PCR and also evaluated a couple of different size selection methods.

 Answer:

We would like to appreciate the reviewer’s corrections and suggestions on our manuscript. We tried to reflect all the valuable comments to the revision.

The introduction could benefit from a better description of the three approaches and the two clean-up methods: specifically, what is the general approach, how they are different, advantages, and disadvantages?

Answer:

We understand the reviewer’s comment that the introduction needs to include the description on the differences of the approaches and methods. We enriched the content of some paragraphs in Introduction to explain the difference of approaches and methods.

Unfortunately, there is no previous report on the direct comparison between PNA and LAN-based methods and between blocking and target specific methods. Therefore, we thought it is better to demonstrate the advantages/disadvantages of each method in Discussion by referring the results obtained in this study.

We realized the need for another clean-up method during the experiment. Since the selection of different clean-up methods is not usually considered, we are afraid that including the description on different clean-up methods in Introduction may confuse the readers. Instead, we introduced the explanation on the two clean-up methods in Materials and Methods.

All the related changed are shown below:

Line 77-85:

LNA-based blockers have been employed in PCR for denaturing gradient gel electropho-resis to inhibit the amplification of plant SSU rRNA genes and to simultaneously detect plant-associated bacteriome [26]. Although both blockers PNA and LNA are expected to bind to the target sequences, their blocking efficiency may differ according to the experimental conditions. In fact, it is known that the binding capacity of DNA and LNA in-creases with a high salt concentration, whereas PNA is not substantially affected by the salt concentration [25,27]. It is unclear how this affects the strength and selectivity of amplification inhibition by artificial nucleic acids, and there are no directly relevant reports.

Line 88-93:

UNonMet-PCR, a method that selectively amplifies the 18S rDNA of non-metazoan organisms, was first reported for the detection of protists in oysters [28] and has since been used for the comprehensive detection of protists in ctenophores, corals, and human stool samples in combination with NGS [29,30]. Because ticks are also metazoans, this method can theoretically be applied to investigate eukaryotes in ticks.Line 202-206:

Then, the index PCR products were purified using AMPure or NucleoMag NGS Clean-up and Size Select (SizeSelect) (Macherey-Nagel, Düren, Germany). The former product is generally used for the purification of > 100 bp DNA fragments, while the latter product can be used to enrich DNA fragments ranging in size from 150–800 bp.

Line 508-521

When the blocker was added to the PCR, the non-specific amplification of the long chain increased (Figure 2). Based on this non-specific amplification without an adaptor sequence, we expected quantitative real-time PCR (qPCR) to accurately measure the concentration; however, the estimated concentration was the same as that of Qubit measurement (data not shown). Thus, we determined that the adapter sequence was attached to the long chain sequence and accurate measurements were impossible, even by qPCR. These non-target sequences should be excluded because they negatively affect concentration adjustment and cluster formation and reduce the final number of acquired reads, even though the origin of these remain unknown. In the present study, the non-specific amplification was successfully mitigated using SizeSelect instead of AMPure, and the diversity of the detected sequences also increased (Figure 3). Unfortunately, it is not clear from this study whether the presence of non-specific amplification is specific to our samples or not. Therefore, it is recommended to evaluate the library size first when using the TickB_PNA and TickB_LNA methods developed in this study.

I would also like to see more biological interpretation of the different results as measured by Shannon’s vs. Faith’s diversity indices. What is the significance of different responses of the blocking methods in these two types of methods?

 Answer:

Thank you for this comment. We also thought that description on the difference between Shannon diversity and Faith's phylogenetic diversity is useful to understand the different responses of the blocking methods. We revised the text as follows:

Line 427-442:

The TickB_PNA, TickB_LNA and UNonMet-PCR strategies all increased the Shan-non diversity and observed ASVs compared with levels in the controls with only universal primers, proving that they can detect a greater diversity of microorganisms (Figure 4). The use of blockers not only increased the alpha diversity, but also increased the frequency of eukaryotes compared with those obtained upon the control and UNonMet-PCR methods (Table 3). This suggests that blockers are appropriate for targeting the entire eukaryotic microbiome of ticks. UNonMet-PCR resulted in a low Faith's PD and this value was not significantly different from that of the control, even though Shannon diversity and observed ASVs were comparable to those obtained with the blockers (Figure 4). Shannon diversity is calculated based on the proportion of species in the total sample, while Faith's PD accounts for the length of the phylogenetic branches. Taken together, the proportion of matching sequences in eukaryotic communities in ticks was lower for UNonMet-PCR-primers than for the TAReuk primer pair, and therefore the phylogeny of sequences detected was weighted to fungi and Apicomplexa (Figure 4) [52]. UNonMet-PCR detects a significantly higher abundance of fungal-derived sequences than the blocker methods and is more suitable for specifically targeting these sequences (Figure 4).

The other area that could benefit from additional clarification is in the methods. Given that many of these approaches are relatively novel and not widely used, it would be helpful for the methods to start with a general introduction to what the goal of each step is. For instance, in section 2.2 Designing Blocker Nucleic Acids, a broad overview of how the blocker works would help explain the design details. Does the blocker nucleic acid get added to the sample before the PCR? During? And in section 2.3 Plasmid preparation, what is the purpose of generating plasmids of tick and apicomplexan 18S DNA?

 Answer:

We agree with the comments and tried to start with a general introduction to the goal of each step. In addition, more detailed information on the addition of blocker nucleic acids is included.

Line 118-120:

We searched for the tick-specific sequences where the blocker nucleic acids can bind, which results in the inhibition of tick DNA amplification while allowing the amplification of other eukaryotic DNA by PCR.

Line 141-143

To evaluate the blocking efficiency of the designed blocker nucleic acids, the plasmids with the insertion of a partial 18S rDNA fragment of a tick (blocking target) and an apicomplexan parasite (amplification target) were prepared.

Line 195-197:

To compare the blocking methods and library purification methods, a total of 17 tick DNA samples listed in Table 1 were subjected to the conventional PCR without blocker (control), PCR with TickB_PNA or TickB_LNA, and UNonMet-PCR.

The discussion is quite thorough but could use some more definitive assessments of the usefulness of the methods tested here. It appears that PNA is effective for incrasing species richness, but then there are some cases where LNA might be better. UNon-Met may also be better in terms of identifying more taxa in more samples. What are the recommendations for each of these methods with respect to different kinds of research questions or objectives. How should researchers decide which of these to use?

 Answer: 

We agree with this comment. We provided the description on the potential of LNA-based method in Discussion. Furthermore, we included the following sentences to explain our recommendation for other researchers:

Line 464-469:

Although many examples of the application of blocker PNAs to amplicon preparation have been reported [20–23], blocker LNAs have not been used for comprehensive analyses by NGS. It is clear that selective amplification inhibition using PNA is useful for amplicon analyses [20–24]; however, for the reasons mentioned above, LNA may also be appropri-ate depending on the conditions, including the blocker sequences and buffers, and more trials are needed to optimise the choice of artificial nucleic acids for blocking.Line 532-537:

We demonstrated comprehensive analysis of eukaryotic organisms in ticks using the blocker and UNonMet-PCR methods for high-throughput sequencing. In addition to known tick-borne protozoa, a wide variety of previously unreported protists were detected. We recommend the use of blocker methods to capture the entire eukaryotic microbiome of ticks. The UNonMet-PCR is also useful in the characterisation of fungal community in ticks.

Also, what is the significance of the SizeSelect method working better than Ampure? How are they different? Is this a general pattern or is it because of the particular amplicons in these samples?

 Answer:

We described the difference of two methods in the Materials and Methods. Unfortunately, it is not clear whether the generation of non-specific amplification is specific to our samples or not. We therefore recommend other researchers to check their library size first and use SizeSelect when necessary. To explain these, the following sentences are added.

Line 202-206:

Then, the index PCR products were purified using AMPure or NucleoMag NGS Clean-up and Size Select (SizeSelect) (Macherey-Nagel, Düren, Germany). The former product is generally used for the purification of > 100 bp DNA fragments, while the latter product can be used to enrich DNA fragments ranging in size from 150–800 bp.

Line 514-521:

These non-target sequences should be excluded because they negatively affect concentra-tion adjustment and cluster formation and reduce the final number of acquired reads, even though the origin of these remain unknown. In the present study, the non-specific amplification was successfully mitigated using SizeSelect instead of AMPure, and the di-versity of the detected sequences also increased (Figure 3). Unfortunately, it is not clear from this study whether the presence of non-specific amplification is specific to our sam-ples or not. Therefore, it is recommended to evaluate the library size first when using the TickB_PNA and TickB_LNA methods developed in this study.

Line 88: needs to be edited for grammar. Suggest: “Herein, we report on the use of new methods…” Although are these new? Several citations are provided that have used the PNA and LNA methods go back to 2012 although several are more recent. Perhaps this whole last paragraph of the introduction could be rephrased to summarize the main goal of the study, to evaluate these three approaches in a tick system.

 Answer: We agree with this suggestion. The manuscript has been revised as follows.

Line 94-99:

The aims of this study were to develop the methods for tick eukaryotic microbiome analysis by designing blocker PNA and LNA, which selectively bind to tick 18S rDNA and inhibit its amplification, and to compare the results between PNA- and LNA-based methods and the UNonMet-PCR. Our results indicated that all the methods effectively suppressed the amplification of tick 18S rDNA and enabled the detection of diverse eu-karyotes in ticks.

Line 176-177: What does it mean that PCR with TickB_PNA or TickB-LNAa… were added with a final concentration? Were these added to the conventional PCR? The blocked PCRS were added to a conventional PCR?  A clearer description of when LNA and PNA blocking occurs would really be helpful.

 Answer:

TickB_PNA or TickB-LNA were added to conventional PCR, which products were subjected to the index PCR using the Nextera XP Kit Illumina Libirary prep Kit. We tried to explain the details of the methods as much as possible. The following sentences are included in the Materials and Methods.

Line 162-169:

PCR was performed in a 25.0 μL reaction mixture containing 12.5 μL of 2× KAPA HiFi HotStart ReadyMix (KAPA Biosystems, Wilmington, MA, USA), 10 μM each primer (TAReuk454FWD1 and TAReuREV3), 2.5 μL of plasmid template (HlP 106 copies/μL, ToP 106 copies/μL or ToP 102 copies/μL), and 0, 1, 5, 10, 50, or 100 pmol TickB_PNA or TickB_LNA.

Line 195-199:

To compare the blocking methods and library purification methods, a total of 17 tick DNA samples listed in Table 1 were subjected to the conventional PCR without blocker (control), PCR with TickB_PNA or TickB_LNA, and UNonMet-PCR. Blockers (TickB_PNA or TickB_LNA) were added at a final concentration of 50 pmol/reaction and PCR reactions were conducted at the same condition as described above.

Reviewer 2 Report

The authors describe methodological approaches to elucidate the eukaryotic microbiome diversity in a context where the host genome would be amplified. Several similar scenarios can benefit from an alternative to host depletion, making the technical focus of the paper relevant and shareable.

Major concern:

1) The manuscript omits the Short Reads Archive (SRA) accession code, making impossible a re-analysis to confirm the major findings. I will be glad to finish the review after this information has been provided.

Minor issues and comments:

1) Abstract and introduction can be improved in explaining the setup, as I initially thought some kind of concurrent use of multiple strategies was being proposed or evaluated (eg: PNA+PLA or PNA+UNonMet primers...). 

2) Paragraph 2.1 is rendered in italic.

3) Paragraph 2.3: I would rephrase the last sentence to put the abbreivated name in brackets after the extended (e.g. fragment of H. lognicornis (HlP) and T. orientalis (ToP)), as looking for abbreviations is faster with brackets.

4) A figure showing the location of the blocking primers and the universal primers position in respect to the tick 18S gene would make it easier to follow the text.

5) Table2 would be clearer aligning the first and second column to the left, possibly using monospaced font for the primer sequence and with a larger second column to avoid breaking primers

6) Paragraph 2.7 reports the incorrect version of DADA2 (if the authors refer to the Qiime2 plugin, they can omit the version); Qiime2 is preferred to QIIME2;

7) Figure 1 misses the panel letters (a, b and c on top of the plots), and Figure 5 should include a description of the three panels in the caption as well. Figure 5 would be probably clearer removing the grouping ellipsis and increasing the samples dots, considering the modest number of samples per plot.

8) Figure 3 would be probably clearer with the title on top instead of vertically placed on the y-axis. Similar colors (eg shades of the same color) for PNA_AMP and LNA_AMP, as well as for PNA_SS and LNA_SS can guide the reader

9) Figure 4 is very hard to read (text too small)

Author Response

Response to reviewers

Reviewer 2:

The authors describe methodological approaches to elucidate the eukaryotic microbiome diversity in a context where the host genome would be amplified. Several similar scenarios can benefit from an alternative to host depletion, making the technical focus of the paper relevant and shareable.

Answer:

We wish to thank the reviewer for corrections and suggestions on our manuscript. We tried to address all the comments and revised the text and figures accordingly.

Major concern:

1) The manuscript omits the Short Reads Archive (SRA) accession code, making impossible a re-analysis to confirm the major findings. I will be glad to finish the review after this information has been provided.

Answer:

Accession number DRA011889 is now provided in the text. We made the data publicly available.

Line 220-222:

Raw sequence data have been deposited in the DNA Data Bank of Japan Sequence Read Archive with the accession number DRA011889.

Line 557-558:

Data Availability Statement: Raw sequence data have been deposited in the DNA Data Bank of Japan (DDBJ) Sequence Read Archive with an accession number of DRA001889.

Minor issues and comments:

1) Abstract and introduction can be improved in explaining the setup, as I initially thought some kind of concurrent use of multiple strategies was being proposed or evaluated (eg: PNA+PLA or PNA+UNonMet primers...). 

Answer:

We apologize for our unclear explanation. Abstract and Introduction are revised as follows:

Abstruct Line 29-33:

In this study, we developed new methods to selectively amplify microeukaryote genes in tick-derived DNA by blocking the amplification of the 18S rRNA gene of ticks using artificial nucleic acids: peptide nucleic acids (PNAs) and locked nucleic acids (LNAs). In addition, another PCR using non-metazoan primers, referred to as UNonMet-PCR, was performed for comparison.

Line 94-97:

The aims of this study were to develop the methods for tick eukaryotic microbiome analysis by designing blocker PNA and LNA, which selectively bind to tick 18S rDNA and inhibit its amplification, and to compare the results between blocker and the UNonMet-PCR methods.

2) Paragraph 2.1 is rendered in italic.

Answer:

The italicized section text has been corrected and only the tick species was italicized. Also, the section titles written in bold were italicised.

3) Paragraph 2.3: I would rephrase the last sentence to put the abbreivated name in brackets after the extended (e.g. fragment of H. lognicornis (HlP) and T. orientalis (ToP)), as looking for abbreviations is faster with brackets.

Answer:

Corrected as suggested.

Line 157-159:

The standard plasmids at the concentrations of 108 copies/µL were obtained for a partial 18S rDNA fragment of H. longicornis (HlP) andT. orientalis (ToP).

4) A figure showing the location of the blocking primers and the universal primers position in respect to the tick 18S gene would make it easier to follow the text.

Answer:

We provided the figure describing the location of primers and blockers as Figure 1.

5) Table2 would be clearer aligning the first and second column to the left, possibly using monospaced font for the primer sequence and with a larger second column to avoid breaking primers

Answer:

We accepted the comments.

6) Paragraph 2.7 reports the incorrect version of DADA2 (if the authors refer to the Qiime2 plugin, they can omit the version); Qiime2 is preferred to QIIME2;

Answer:

We accepted the comments.

7) Figure 1 misses the panel letters (a, b and c on top of the plots), and Figure 5 should include a description of the three panels in the caption as well. Figure 5 would be probably clearer removing the grouping ellipsis and increasing the samples dots, considering the modest number of samples per plot.

Answer:

We accepted the comments on Figure 1 (Figure 2 in revision). The description on the three panels of Figure 5 (Figure 6 in revision) is now provided in the caption. We agree with the comments on the dot size and made them bigger for better view. We believe that it is better to keep the grouping because they indicates the statistical prediction of the range of plots for each group. We are afraid that the removing ellipsis may lead to misunderstanding of the data. For instance, the red and blue groups in “c” have overlapping ellipsis, indicating that the overlapping between tick species is not rejected. However, without ellipsis, they might appear to be completely separated.

8) Figure 3 would be probably clearer with the title on top instead of vertically placed on the y-axis. Similar colors (eg shades of the same color) for PNA_AMP and LNA_AMP, as well as for PNA_SS and LNA_SS can guide the readers

Answer:

We accepted the comments.

9) Figure 4 is very hard to read (text too small)

Answer:

We appreciate this comment. We tried to enlarge the letters embedded in the figure as much as possible.

Round 2

Reviewer 1 Report

I am satisfied with the authors' responses to my first round of comments and suggestions.

Author Response

I am satisfied with the authors' responses to my first round of comments and suggestions.

Answer:

We would like to thank you for your prompt and precise suggestions.

Reviewer 2 Report

The authors improved the manuscript, and added the necessary SRA link (that is correct in the text, but wrong in the footnotes, where is reported as DRA001889).

The manuscript has been substantially revisited with all the comments addressed; I'd like to acknowledge the appropriate and timely revisions.

I still have a concern on how the results are presented, that became more evident after I performed a quick and partial QC on some samples. 

To clarify the part of QC I made, I randomly selected these samples: 

DRR288058
DRR288059
DRR288060
DRR288061
DRR288064
DRR288065
DRR288066
DRR288067
DRR288070
DRR288071
DRR288072
DRR288073
DRR288076
DRR288077
DRR288078
DRR288079
DRR288082
DRR288084
DRR288085
DRR288088
DRR288089
DRR288090
DRR288091
DRR288094
DRR288095
DRR288096
DRR288097
DRR288100
DRR288101
DRR288102
DRR288103
DRR288106
DRR288107
DRR288165

The top 10 features I found were:

>repseq1 Ixodes pavlovskyi 
CCAGCAGCCGCGGTAATTCCAGCTCCAATAGCGTATACTAAAGTTGCTGCGGTTAAAAAGCTCGTAGTTGGATCTCAGTTACAGGCGGGAAGTGCGTGGACACCACGTTACGGCCCGTGCTGAACATCATGCCTGTCGTGGCTTGGTTCCCTTCATCAGGTGCCTTGCCTTGGCCGGCGCGTTTACTTTGAAAAAATTAGAGTGCTCAACGCAGGCGATTCGCCTGAATAACAGTGCATGGAATAATAGAACAAGATCCTGTTTCTGTTCTGTTGGTTTTTGGAATACAGGATAATGATTAAGAGGGACAGACGGGGGCATTCGTATTGCGGCGCTAGAGGTGAAATTCTTGGACCGTCGCAAGACGAACTACTGCGAAAGCATTTGCCAAGAATGTTTTCATTGATCAAGAACGAAAGT
>repseq2 Haemaphysalis longicornis
CCAGCAGCCGCGGTAATTCCAGCTCCAATAGCGTATACTAAAGCTGCTGCGGTTAAAAAGCTCGTAGTTGGATCTCAGTTCCAGACGAGTAGTGCATCTACCCGATGCGACGGCTCGGACTGAACATCATGCCGGTCGTTTCTTGGTGCACTTCATTGTGTGCCTCGAGATGGCCGGTGCTTTTACTTTGAAAAAATTAGAGTGCTCAACGCAGGCGAGTCGCCTGAATATTCCTTGCATGGAATAATAGAACAAGACCTCGTTTCTGTTCTGTTGGTTTTTGGAATACGAGGTAATGATTAAGAGGGACAGACGGGGGCATTCGTATTGCGGCGCTAGAGGTGAAATTCTTGGACCGTCGCAAGACGAACTACTGCGAAAGCATTTGCCAAGAATGTTTTCTTTGATCAAGAACGAAAGT
>repseq3 Ixodes pavlovskyi
CCAGCACCCGCGGTAATTCCAGCTCCAATAGCGTATACTAAAGTTGCTGCGGTTAAAAAGCTCGTAGTTGGATCTCAGTTACAGGCGGGAAGTGCGTGGACACCACGTTACGGCCCGTGCTGAACATCATGCCTGTCGTGGCTTGGTTCCCTTCATCAGGTGCCTTGCCTTGGCCGGCGCGTTTACTTTGAAAAAATTAGAGTGCTCAACGCAGGCGATTCGCCTGAATAACAGTGCATGGAATAATAGAACAAGATCCTGTTTCTGTTCTGTTGGTTTTTGGAATACAGGATAATGATTAAGAGGGACAGACGGGGGCATTCGTATTGCGGCGCTAGAGGTGAAATTCTTGGACCGTCGCAAGACGAACTACTGCGAAAGCATTTGCCAAGAATGTTTTCATTGATCAAGAACGAAAGT
>repseq4 Amblyomma hebraeum
CCAGCAGCCGCGGTAATTCCAGCTCCAATAGCGTATACTAAAGCTGCTGCGGTTAAAAAGCTCGTAGTTGGATCTCAGTTCCAGACGAGTAGTGCATCTACCCGATGCGACGGCTCGGACTGAACATCATGCCGGTCCTCTCTTGGTGCCCTTCATTGGTGCGTCTCGAGGTGGCCCGCGCTTTTACTTTGAAAAAATTAGAGTGCTCAACGCAGGCGAGTCGCCTGAATAAACTTGCATGGAATAATAGAACAAGAGCCCGTTTCTGTTCTGTTGGTTTTTGGAATACGGGCTAATGATTAAGAGGGACAGACGGGGGCATTCGTATTGCGGCGCTAGAGGTGAAATTCTTGGACCGTCGCAAGACGAACTACTGCGAAAGCATTTGCCAAGAATGTTTTCTTTGATCAAGAACGAAAGT
>repseq5 Haemaphysalis longicornis 
CCAGCACCCGCGGTAATTCCAGCTCCAATAGCGTATACTAAAGCTGCTGCGGTTAAAAAGCTCGTAGTTGGATCTCAGTTCCAGACGAGTAGTGCATCTACCCGATGCGACGGCTCGGACTGAACATCATGCCGGTCGTTTCTTGGTGCACTTCATTGTGTGCCTCGAGATGGCCGGTGCTTTTACTTTGAAAAAATTAGAGTGCTCAACGCAGGCGAGTCGCCTGAATATTCCTTGCATGGAATAATAGAACAAGACCTCGTTTCTGTTCTGTTGGTTTTTGGAATACGAGGTAATGATTAAGAGGGACAGACGGGGGCATTCGTATTGCGGCGCTAGAGGTGAAATTCTTGGACCGTCGCAAGACGAACTACTGCGAAAGCATTTGCCAAGAATGTTTTCTTTGATCAAGAACGAAAGT
>repseq6 Ixodes pavlovskyi
CCAGCAGCTGCGGTAATTCCAGCTCCAATAGCGTATACTAAAGTTGCTGCGGTTAAAAAGCTCGTAGTTGGATCTCAGTTACAGGCGGGAAGTGCGTGGACACCACGTTACGGCCCGTGCTGAACATCATGCCTGTCGTGGCTTGGTTCCCTTCATCAGGTGCCTTGCCTTGGCCGGCGCGTTTACTTTGAAAAAATTAGAGTGCTCAACGCAGGCGATTCGCCTGAATAACAGTGCATGGAATAATAGAACAAGATCCTGTTTCTGTTCTGTTGGTTTTTGGAATACAGGATAATGATTAAGAGGGACAGACGGGGGCATTCGTATTGCGGCGCTAGAGGTGAAATTCTTGGACCGTCGCAAGACGAACTACTGCGAAAGCATTTGCCAAGAATGTTTTCATTGATCAAGAACGAAAGT
>repseq7 Amblyomma hebraeum
CCAGCACCCGCGGTAATTCCAGCTCCAATAGCGTATACTAAAGCTGCTGCGGTTAAAAAGCTCGTAGTTGGATCTCAGTTCCAGACGAGTAGTGCATCTACCCGATGCGACGGCTCGGACTGAACATCATGCCGGTCCTCTCTTGGTGCCCTTCATTGGTGCGTCTCGAGGTGGCCCGCGCTTTTACTTTGAAAAAATTAGAGTGCTCAACGCAGGCGAGTCGCCTGAATAAACTTGCATGGAATAATAGAACAAGAGCCCGTTTCTGTTCTGTTGGTTTTTGGAATACGGGCTAATGATTAAGAGGGACAGACGGGGGCATTCGTATTGCGGCGCTAGAGGTGAAATTCTTGGACCGTCGCAAGACGAACTACTGCGAAAGCATTTGCCAAGAATGTTTTCTTTGATCAAGAACGAAAGT
>repseq8 Haemaphysalis longicornis
CCAGCAGCTGCGGTAATTCCAGCTCCAATAGCGTATACTAAAGCTGCTGCGGTTAAAAAGCTCGTAGTTGGATCTCAGTTCCAGACGAGTAGTGCATCTACCCGATGCGACGGCTCGGACTGAACATCATGCCGGTCGTTTCTTGGTGCACTTCATTGTGTGCCTCGAGATGGCCGGTGCTTTTACTTTGAAAAAATTAGAGTGCTCAACGCAGGCGAGTCGCCTGAATATTCCTTGCATGGAATAATAGAACAAGACCTCGTTTCTGTTCTGTTGGTTTTTGGAATACGAGGTAATGATTAAGAGGGACAGACGGGGGCATTCGTATTGCGGCGCTAGAGGTGAAATTCTTGGACCGTCGCAAGACGAACTACTGCGAAAGCATTTGCCAAGAATGTTTTCTTTGATCAAGAACGAAAGT
>repseq9 Dermacentor niveus
CCAGCAGCCGCGGTAATTCCAGCTCCAATAGCGTATACTAAAGCTGCTGCGGTTAAAAAGCTCGTAGTTGGATCTCAGTTCCAGACGAGTAGTGCATCTACCCGATGCGACGGCTCGGACTGAACATCATGCCGGTCCTTTCTTGGTGCACTTCATTGTGTGCCTCGAGAAGGCCGGTGCTTTTACTTTGAAAAAATTAGAGTGCTCAACGCAGGCGAGTCGCCTGAATAAACTTGCATGGAATAATAGAACAAGACCTCGTTTCTGTTCTGTTGGTTTTTGGAATACGAGGTAATGATTAAGAGGGACAGACGGGGGCATTCGTATTGCGGCGCTAGAGGTGAAATTCTTGGACCGTCGCAAGACGAACTACTGCGAAAGCATTTGCCAAGAATGTTTTCTTTGATCAAGAACGAAAGT
>repseq10 Amblyomma hebraeum
CCAGCAGCTGCGGTAATTCCAGCTCCAATAGCGTATACTAAAGCTGCTGCGGTTAAAAAGCTCGTAGTTGGATCTCAGTTCCAGACGAGTAGTGCATCTACCCGATGCGACGGCTCGGACTGAACATCATGCCGGTCCTCTCTTGGTGCCCTTCATTGGTGCGTCTCGAGGTGGCCCGCGCTTTTACTTTGAAAAAATTAGAGTGCTCAACGCAGGCGAGTCGCCTGAATAAACTTGCATGGAATAATAGAACAAGAGCCCGTTTCTGTTCTGTTGGTTTTTGGAATACGGGCTAATGATTAAGAGGGACAGACGGGGGCATTCGTATTGCGGCGCTAGAGGTGAAATTCTTGGACCGTCGCAAGACGAACTACTGCGAAAGCATTTGCCAAGAATGTTTTCTTTGATCAAGAACGAAAGT

In the selected samples, the first representative sequences account for 70% of the total reads, and as far as I understand they map against tick species. I have two comments on this part of the paper.

1) My results is in my opinion are in agreement with Table 3, but at the same time shows how uninformative is Table 3 for the reader, and should be complemented with a more detailed chart depicting the ratio of host reads where the distribution of values is appreciable (i.e.  boxplots with dots as done with Figure 4 or violin plots). 

2) It's easy to suspect that the variability observed in % of host reads is more due to the sample preparation than to the library, as the Control samples in Table 3 can reach a maximum of 80% of non host reads. I wonder if the results should be provided also in terms of change with the control, so given a specific DNA extraction, comparing the performance of control and the other methods, showing the enrichment normalized with each individual control. 

In practical terms, this part is pivotal in guiding the reader in evaluating if the host suppression methods are worth being tried in their case.

Author Response

The authors improved the manuscript, and added the necessary SRA link (that is correct in the text, but wrong in the footnotes, where is reported as DRA001889).

Answer:

We would like to appreciate the reviewer’s corrections and suggestions on the manuscript. We apologize for the wrong information in the footnote. Now the number is corrected to DRA011889.

The manuscript has been substantially revisited with all the comments addressed; I'd like to acknowledge the appropriate and timely revisions.

I still have a concern on how the results are presented, that became more evident after I performed a quick and partial QC on some samples. 

To clarify the part of QC I made, I randomly selected these samples: 

DRR288058
DRR288059
DRR288060
DRR288061
DRR288064
DRR288065
DRR288066
DRR288067
DRR288070
DRR288071
DRR288072
DRR288073
DRR288076
DRR288077
DRR288078
DRR288079
DRR288082
DRR288084
DRR288085
DRR288088
DRR288089
DRR288090
DRR288091
DRR288094
DRR288095
DRR288096
DRR288097
DRR288100
DRR288101
DRR288102
DRR288103
DRR288106
DRR288107
DRR288165

The top 10 features I found were:

>repseq1 Ixodes pavlovskyi 
CCAGCAGCCGCGGTAATTCCAGCTCCAATAGCGTATACTAAAGTTGCTGCGGTTAAAAAGCTCGTAGTTGGATCTCAGTTACAGGCGGGAAGTGCGTGGACACCACGTTACGGCCCGTGCTGAACATCATGCCTGTCGTGGCTTGGTTCCCTTCATCAGGTGCCTTGCCTTGGCCGGCGCGTTTACTTTGAAAAAATTAGAGTGCTCAACGCAGGCGATTCGCCTGAATAACAGTGCATGGAATAATAGAACAAGATCCTGTTTCTGTTCTGTTGGTTTTTGGAATACAGGATAATGATTAAGAGGGACAGACGGGGGCATTCGTATTGCGGCGCTAGAGGTGAAATTCTTGGACCGTCGCAAGACGAACTACTGCGAAAGCATTTGCCAAGAATGTTTTCATTGATCAAGAACGAAAGT
>repseq2 Haemaphysalis longicornis
CCAGCAGCCGCGGTAATTCCAGCTCCAATAGCGTATACTAAAGCTGCTGCGGTTAAAAAGCTCGTAGTTGGATCTCAGTTCCAGACGAGTAGTGCATCTACCCGATGCGACGGCTCGGACTGAACATCATGCCGGTCGTTTCTTGGTGCACTTCATTGTGTGCCTCGAGATGGCCGGTGCTTTTACTTTGAAAAAATTAGAGTGCTCAACGCAGGCGAGTCGCCTGAATATTCCTTGCATGGAATAATAGAACAAGACCTCGTTTCTGTTCTGTTGGTTTTTGGAATACGAGGTAATGATTAAGAGGGACAGACGGGGGCATTCGTATTGCGGCGCTAGAGGTGAAATTCTTGGACCGTCGCAAGACGAACTACTGCGAAAGCATTTGCCAAGAATGTTTTCTTTGATCAAGAACGAAAGT
>repseq3 Ixodes pavlovskyi
CCAGCACCCGCGGTAATTCCAGCTCCAATAGCGTATACTAAAGTTGCTGCGGTTAAAAAGCTCGTAGTTGGATCTCAGTTACAGGCGGGAAGTGCGTGGACACCACGTTACGGCCCGTGCTGAACATCATGCCTGTCGTGGCTTGGTTCCCTTCATCAGGTGCCTTGCCTTGGCCGGCGCGTTTACTTTGAAAAAATTAGAGTGCTCAACGCAGGCGATTCGCCTGAATAACAGTGCATGGAATAATAGAACAAGATCCTGTTTCTGTTCTGTTGGTTTTTGGAATACAGGATAATGATTAAGAGGGACAGACGGGGGCATTCGTATTGCGGCGCTAGAGGTGAAATTCTTGGACCGTCGCAAGACGAACTACTGCGAAAGCATTTGCCAAGAATGTTTTCATTGATCAAGAACGAAAGT
>repseq4 Amblyomma hebraeum
CCAGCAGCCGCGGTAATTCCAGCTCCAATAGCGTATACTAAAGCTGCTGCGGTTAAAAAGCTCGTAGTTGGATCTCAGTTCCAGACGAGTAGTGCATCTACCCGATGCGACGGCTCGGACTGAACATCATGCCGGTCCTCTCTTGGTGCCCTTCATTGGTGCGTCTCGAGGTGGCCCGCGCTTTTACTTTGAAAAAATTAGAGTGCTCAACGCAGGCGAGTCGCCTGAATAAACTTGCATGGAATAATAGAACAAGAGCCCGTTTCTGTTCTGTTGGTTTTTGGAATACGGGCTAATGATTAAGAGGGACAGACGGGGGCATTCGTATTGCGGCGCTAGAGGTGAAATTCTTGGACCGTCGCAAGACGAACTACTGCGAAAGCATTTGCCAAGAATGTTTTCTTTGATCAAGAACGAAAGT
>repseq5 Haemaphysalis longicornis 
CCAGCACCCGCGGTAATTCCAGCTCCAATAGCGTATACTAAAGCTGCTGCGGTTAAAAAGCTCGTAGTTGGATCTCAGTTCCAGACGAGTAGTGCATCTACCCGATGCGACGGCTCGGACTGAACATCATGCCGGTCGTTTCTTGGTGCACTTCATTGTGTGCCTCGAGATGGCCGGTGCTTTTACTTTGAAAAAATTAGAGTGCTCAACGCAGGCGAGTCGCCTGAATATTCCTTGCATGGAATAATAGAACAAGACCTCGTTTCTGTTCTGTTGGTTTTTGGAATACGAGGTAATGATTAAGAGGGACAGACGGGGGCATTCGTATTGCGGCGCTAGAGGTGAAATTCTTGGACCGTCGCAAGACGAACTACTGCGAAAGCATTTGCCAAGAATGTTTTCTTTGATCAAGAACGAAAGT
>repseq6 Ixodes pavlovskyi
CCAGCAGCTGCGGTAATTCCAGCTCCAATAGCGTATACTAAAGTTGCTGCGGTTAAAAAGCTCGTAGTTGGATCTCAGTTACAGGCGGGAAGTGCGTGGACACCACGTTACGGCCCGTGCTGAACATCATGCCTGTCGTGGCTTGGTTCCCTTCATCAGGTGCCTTGCCTTGGCCGGCGCGTTTACTTTGAAAAAATTAGAGTGCTCAACGCAGGCGATTCGCCTGAATAACAGTGCATGGAATAATAGAACAAGATCCTGTTTCTGTTCTGTTGGTTTTTGGAATACAGGATAATGATTAAGAGGGACAGACGGGGGCATTCGTATTGCGGCGCTAGAGGTGAAATTCTTGGACCGTCGCAAGACGAACTACTGCGAAAGCATTTGCCAAGAATGTTTTCATTGATCAAGAACGAAAGT
>repseq7 Amblyomma hebraeum
CCAGCACCCGCGGTAATTCCAGCTCCAATAGCGTATACTAAAGCTGCTGCGGTTAAAAAGCTCGTAGTTGGATCTCAGTTCCAGACGAGTAGTGCATCTACCCGATGCGACGGCTCGGACTGAACATCATGCCGGTCCTCTCTTGGTGCCCTTCATTGGTGCGTCTCGAGGTGGCCCGCGCTTTTACTTTGAAAAAATTAGAGTGCTCAACGCAGGCGAGTCGCCTGAATAAACTTGCATGGAATAATAGAACAAGAGCCCGTTTCTGTTCTGTTGGTTTTTGGAATACGGGCTAATGATTAAGAGGGACAGACGGGGGCATTCGTATTGCGGCGCTAGAGGTGAAATTCTTGGACCGTCGCAAGACGAACTACTGCGAAAGCATTTGCCAAGAATGTTTTCTTTGATCAAGAACGAAAGT
>repseq8 Haemaphysalis longicornis
CCAGCAGCTGCGGTAATTCCAGCTCCAATAGCGTATACTAAAGCTGCTGCGGTTAAAAAGCTCGTAGTTGGATCTCAGTTCCAGACGAGTAGTGCATCTACCCGATGCGACGGCTCGGACTGAACATCATGCCGGTCGTTTCTTGGTGCACTTCATTGTGTGCCTCGAGATGGCCGGTGCTTTTACTTTGAAAAAATTAGAGTGCTCAACGCAGGCGAGTCGCCTGAATATTCCTTGCATGGAATAATAGAACAAGACCTCGTTTCTGTTCTGTTGGTTTTTGGAATACGAGGTAATGATTAAGAGGGACAGACGGGGGCATTCGTATTGCGGCGCTAGAGGTGAAATTCTTGGACCGTCGCAAGACGAACTACTGCGAAAGCATTTGCCAAGAATGTTTTCTTTGATCAAGAACGAAAGT
>repseq9 Dermacentor niveus
CCAGCAGCCGCGGTAATTCCAGCTCCAATAGCGTATACTAAAGCTGCTGCGGTTAAAAAGCTCGTAGTTGGATCTCAGTTCCAGACGAGTAGTGCATCTACCCGATGCGACGGCTCGGACTGAACATCATGCCGGTCCTTTCTTGGTGCACTTCATTGTGTGCCTCGAGAAGGCCGGTGCTTTTACTTTGAAAAAATTAGAGTGCTCAACGCAGGCGAGTCGCCTGAATAAACTTGCATGGAATAATAGAACAAGACCTCGTTTCTGTTCTGTTGGTTTTTGGAATACGAGGTAATGATTAAGAGGGACAGACGGGGGCATTCGTATTGCGGCGCTAGAGGTGAAATTCTTGGACCGTCGCAAGACGAACTACTGCGAAAGCATTTGCCAAGAATGTTTTCTTTGATCAAGAACGAAAGT
>repseq10 Amblyomma hebraeum
CCAGCAGCTGCGGTAATTCCAGCTCCAATAGCGTATACTAAAGCTGCTGCGGTTAAAAAGCTCGTAGTTGGATCTCAGTTCCAGACGAGTAGTGCATCTACCCGATGCGACGGCTCGGACTGAACATCATGCCGGTCCTCTCTTGGTGCCCTTCATTGGTGCGTCTCGAGGTGGCCCGCGCTTTTACTTTGAAAAAATTAGAGTGCTCAACGCAGGCGAGTCGCCTGAATAAACTTGCATGGAATAATAGAACAAGAGCCCGTTTCTGTTCTGTTGGTTTTTGGAATACGGGCTAATGATTAAGAGGGACAGACGGGGGCATTCGTATTGCGGCGCTAGAGGTGAAATTCTTGGACCGTCGCAAGACGAACTACTGCGAAAGCATTTGCCAAGAATGTTTTCTTTGATCAAGAACGAAAGT

In the selected samples, the first representative sequences account for 70% of the total reads, and as far as I understand they map against tick species. I have two comments on this part of the paper.

1) My results is in my opinion are in agreement with Table 3, but at the same time shows how uninformative is Table 3 for the reader, and should be complemented with a more detailed chart depicting the ratio of host reads where the distribution of values is appreciable (i.e.  boxplots with dots as done with Figure 4 or violin plots). 

2) It's easy to suspect that the variability observed in % of host reads is more due to the sample preparation than to the library, as the Control samples in Table 3 can reach a maximum of 80% of non host reads. I wonder if the results should be provided also in terms of change with the control, so given a specific DNA extraction, comparing the performance of control and the other methods, showing the enrichment normalized with each individual control. 

In practical terms, this part is pivotal in guiding the reader in evaluating if the host suppression methods are worth being tried in their case.

Answer:

First of all, we deeply thank the reviewer for taking his/her time to reanalyse our data and making very useful suggestions. As the reviewer’s suggested, showing enrichment by normalizing with each individual control greatly improved the interpretation of the data. In order to show this enrichment effect, we calculated the “non-tick read enrichment value” by dividing the proportion of non-tick reads of each method by that obtained for each individual control. We also added a boxplot (Figure 4) showing enrichment value of non-tick reads instead of Table 3 (table number before revision round 2). The details of values and number of reads are provided as Table S2. In addition, Figure S3 was added to visualize the relative abundance of eukaryotic taxa detected in individual samples which support the discussion about the reason for variation of increased ratio among samples.

All the changes related to the comments are as follows:

Lines 311-326:

The proportions of non-tick reads to all eukaryote reads were significantly higher for PCR with blockers and UNonMet-PCR than those for control (p < 0.01) (and the proportions of tick reads were lower) (Figure 4; Table S2). The median proportions of non-tick reads to all eukaryote reads were 0.03% for the control method, while higher median values 3.24%, 5.75%, 1.53%, 3.18%, and 1.37% were obtained for TickB_PNA with AMPure, TickB_PNA with SizeSelect, TickB_LNA with AMPure, TickB_LNA with Size Select, and UNonMet-PCR, respectively (Table S2). The tick ID 467 showed the high proportion (81.14%) of non-tick reads even with control method. Figure 4 indicates the non-tick read enrichment value calculated by dividing the proportion of non-tick reads of each method by that obtained for each individual control. The median non-tick read enrichment values for TickB_PNA with AMPure, TickB_PNA with SizeSelect, TickB_LNA with AMPure, TickB_LNA with Size Select, and UNonMet-PCR were 60.3, 103.8, 19.6, 32.9, and 35.2 -folds, respectively (Table S2). Upon using PNA-based methods, all samples except for tick ID 467 showed more than 10-fold enrichment (ranging between 12.8 and 705.0 folds). The decreased proportion of non-tick reads compared to control was only observed in the tick IDs 66 and 2876 upon using UNonMet-PCR.

Lines 328-334:

Figure 4. Comparison of non-tick read enrichment values calculated for each method. The non-tick read enrichment value was calculated by dividing the proportion of non-tick reads of each method by that obtained for each individual control. Universal primer set; TAReuk454FWD1 and TAReukREV3 (Table 2) were used for PCRs of the control, TickB_PNA (PNA) and TickB_LNA (LNA). The primer sets; EUK581F and EUK1134R were used for the first PCR, and E572F and E1009R were used for the second PCR for UNonMet-PCR(UNM). The purification methods were AMPure (AMP) or SizeSelect (SS). Tick ID 3611 was filtered out because only tick reads were detected in the control method (Table S2). The y-axis is given as logarithmic axis. Outlier plots were indicated by additional dots in grey.

Discussion.

Lines 442-451:

It is worth mentioning that one sample (tick ID 467) showed extremely high proportion (81.14%) of non-tick reads with the control method (Table S2). Since all the sample processing procedures including DNA extraction are the same, one possible reason is the presence of high abundance of fungi in the tested tick due to environmental contamination or infection to the tick (Figure S3). Another possible explanation is the presence of nucleotide mismatches at the primer binding sites of the 18S rDNA sequences of the tested tick, resulting in poor amplification of tick DNA. Nonetheless, the fact that the proportion of non-tick reads was greatly increased in all other samples by blocker and UNonMet-PCR indicates that these techniques are useful for analyses of the eukaryotic microbiome in ticks.